# First Patient-to-Patient Intrahospital Transmission of Clade I *Candida auris* in France Revealed after a Two-Month Incubation Period

Alexandre Alanio,[a,b,c] Hannah Marie Snell,[d] Camille Cordier,[b] Marie Desnos-Olivier,[a] Sarah Dellière,[a,b,c] Nesrine Aissaoui,[b] Aude Sturny-Leclère,[a] Elodie Da Silva,[b] Cyril Eblé,[d] Martine Rouveau,[d] Micheline Thégat,[d] Widad Zebiche,[d] Matthieu Lafaurie,[e] Blandine Denis,[e] Sophie Touratier,[f] Mourad Benyamina,[g,h] Emmanuel Dudoignon,[g,h] Samia Hamane,[b] Christina A. Cuomo,[d] François Dépret[c,g,h,i,j,k]

aInstitut Pasteur, Université Paris Cité, CNRS, Unité de Mycologie Moléculaire, Centre National de Référence Mycoses Invasives et Antifongiques, UMR2000, Paris, France
bLaboratoire de parasitologie-mycologie, Assistance Publique-Hôpitaux de Paris, Hôpital Saint-Louis, Paris, France
cUniversité Paris Cité, Paris, France
dBroad Institute of MIT and Harvard, Cambridge, Massachusetts, USA
eEquipe Opérationnelle d'Hygiène, Groupe Hospitalier Lariboisière, Saint-Louis, Fernand Widal, Assistance Publique-Hôpitaux de Paris, Paris, France
fService de maladies infectieuses et tropicales, Groupe Hospitalier Lariboisière, Saint-Louis, Fernand Widal, Assistance Publique-Hôpitaux de Paris, Paris, France
gPharmacie centrale, Groupe Hospitalier Lariboisière, Saint-Louis, Fernand Widal, Assistance Publique-Hôpitaux de Paris, Paris, France
hDépartement d'anesthésie réanimation, réanimation chirurgicale et centre de traitement des brûlés, Groupe Hospitalier Lariboisière, Saint-Louis, Fernand Widal, Assistance Publique-Hôpitaux de Paris, Paris, France
iFHU PROMICE, Paris, France
jINSERM UMR-942, Paris, France
kRéseau INI-CRCT, Nancy, France

**ABSTRACT** *Candida auris* is a recently described emerging pathogen in hospital settings. Five genetic clades have been delineated, with each clade being isolated from specific geographic regions. We here describe the first transmission between 2 patients (P0 and P1) of a clade I *C. auris* strain imported into our burn intensive care unit from the Middle East. The strains have been investigated with whole-genome sequencing, which validated the high similarity of the genomes between isolates from P0 and P1. We repeatedly screened the two patients and contact patients (i.e., other patients present in the same hospital ward at the time of the first positive sample from P0 or P1; $n = 49$; 268 tests) with fungal culture and a *C. auris*-specific quantitative PCR assay to assess transmission patterns. We observed that P1 developed *C. auris* colonization between 41 and 61 days after potential exposure to P0 contamination, despite three negative screening tests as recommended by our national authorities. This study illustrates that transmission of *C. auris* between patients can lead to long-term incubation times before the detection of colonization. The recommended screening strategy may not be optimal and should be improved in the light of our findings.

**IMPORTANCE** While large outbreaks of *C. auris* in hospital settings have been described, few clear cases of direct transmission have been documented. We here investigated the transmission of *C. auris* clade I between two patients with a 41- to 61-day delay between exposure and the development of colonization. This may lead to changes in the recommendations concerning treatment of *C. auris* cases, as an incubation period of this length is one of the first to be reported.

**KEYWORDS** *Candida auris*, burn, ICU, qPCR, outbreak, transmission, whole genome sequencing, SNPs, incubation

Address correspondence to Alexandre Alanio, alexandre.alanio@pasteur.fr.

The authors declare no conflict of interest.

*C*andida auris is an emerging fungal ascomycetes species, first described in 2009, that has spread worldwide recently (1). Isolates from all over the world have been investigated, and 5 main genetic clades have been identified to date (2, 3). The origin and emergence of this species is not well understood, but environmental and climatic changes have been brought to the fore (4, 5). The resistance rates of *C. auris* vary, reaching 90% resistance to fluconazole in many countries and up to 5% echinocandin resistance in the United States (6), with 40% multidrug-resistant isolates in some specific geographic areas (7).

Most importantly, this organism has been involved in outbreaks in hospital settings (1, 6). The mean time between admission and onset of candidemia due to *C. auris* has been described as 24 days, based on early reports without prospective microbiological follow-up (8). Upon specific prospective screening, the time from admission to *C. auris* detection was estimated at 4 days in an intensive care unit (ICU) (9), and transmission between patients was 16 days apart, but details on delays of transmission between patients were lacking in most other reports (10–12). Shared material between patients has been demonstrated to be the source of multiple contaminations (10). Transmission to patients through health care personnel is also plausible (9, 12). Several recommendations regarding *C. auris* transmission control have been released over the last few years from the Centers for Disease Control and Prevention (CDC; Atlanta, GA, USA) (13), the European CDC (ECDC) (14), Public Health England (15), and the public health authorities in France (16).

We here describe the details of the transmission of *C. auris* between two burn patients, including the use of a *C. auris* quantitative PCR (qPCR) to screen contact patients (i.e., other patients present in the same hospital unit at the time a patient tested positive for *C. auris*), patients' environments, and whole-genome sequencing to support the clonal transmission of an isolate between patients P0 and P1.

## RESULTS

**Investigation of a *C. auris* candidemia case in a burn ICU.** A 67-year-old woman was admitted to our burn intensive care unit (ICU) on 21 January 2021 for the management of severe burns (>45% of the body surface area) following a domestic accident. The patient was transferred from Abu Dhabi 10 days after the onset of the burn. In our burn ICU, the patient was placed in a single room equipped with dedicated air treatment and a decontamination room but was not tested for *C. auris* colonization upon admission. Dressing of the lesions was performed every 48 h, and topical treatment was adjusted to known skin bacterial colonizers. Upon arrival, as required by internal policies, the patient was placed under contact-protective isolation to avoid potential dissemination of multidrug-resistant bacterial isolates.

Initial superficial skin mycological cultures (swabs) showed skin, respiratory tract, and gut colonization with *Candida glabrata* and *Candida albicans*. A dialysis catheter from this patient (P0) was reported positive with *C. auris* in culture, and the mycology lab was notified at this point. Protective measures against dissemination in the ICU were reinforced but without nurse dedication. In the meantime, *C. auris* was found in blood 7 days after a subculture of a mixed candidemia of *C. glabrata* and *C. parapsilosis*. Individual colonies were identified with Bruker (v4) and Vitek (v3.2) matrix-assisted laser desorption ionization–time of flight mass spectrometry (MALDI-TOF MS). The patient was treated with caspofungin at 70 mg/day. Mycological investigations on retrospective stored specimens showed that the first *C. auris*-positive specimens were on the flank (based on culture) and on a venous catheter (based on qPCR), on the day of the blood culture. Etest and EUCAST antifungal susceptibility testing on different isolates from P0 showed high MICs only to fluconazole (see Table S1 in the supplemental material). The three isolates of P0 investigated belonged to clade I, with only ~960 single-nucleotide polymorphisms (SNPs) different from the clade I reference strain B8441, while more than 35,000 SNPs were observed with clades II, III, and IV (~51,072, ~35,815, and ~130,208 SNPs different, respectively) (Table 1). The maximum SNPs

**TABLE 1** Number of SNP differences between investigated strains[a]

| Strain | No. of SNP differences between strains | | | | | | | | | | |
|---|---|---|---|---|---|---|---|---|---|---|---|
| | CNRMA15-337 | CNRMA17-614 | **CNRMA21-252 P1** | **CNRMA21-86 P0** | **CNRMA21-87 P0** | **CNRMA21-88 P0** | CNRMA7-797 | B11220 (clade II) | B11221 (clade III) | B11243 (clade IV) | Reference B8441 (clade I) |
| CNRMA15-337 | | 143 | 75 | 68 | 65 | 70 | 82 | 57,557 | 40,079 | 147,020 | 949 |
| CNRMA17-614 | 143 | | 121 | 116 | 117 | 115 | 153 | 57,009 | 39,512 | 145,320 | 1,022 |
| **CNRMA21-252 P1** | 75 | 121 | | 12 | 6 | 11 | 97 | 48,080 | 33,718 | 123,175 | 970 |
| **CNRMA21-86 P0** | 68 | 116 | 12 | | 6 | 6 | 97 | 49,600 | 35,323 | 126,690 | 962 |
| **CNRMA21-87 P0** | 65 | 117 | 6 | 6 | | 5 | 95 | 50,791 | 35,466 | 130,297 | 958 |
| **CNRMA21-88 P0** | 70 | 115 | 11 | 6 | 5 | | 95 | 52,826 | 36,656 | 13,3636 | 961 |
| CNRMA7-797 | 82 | 153 | 97 | 97 | 95 | 95 | | 55,323 | 38,848 | 141,979 | 922 |
| B11220 (clade II) | 57,557 | 57,009 | 48,080 | 49,600 | 50,791 | 52,826 | 55,323 | | 62,878 | 163,921 | 64,316 |
| B11221 (clade III) | 40,079 | 39,512 | 33,718 | 35,323 | 35,466 | 36,656 | 38,848 | 62878 | | 165,398 | 44,605 |
| B11243 (clade IV) | 147,020 | 145,320 | 123,175 | 126,690 | 130,297 | 13,3636 | 141,979 | 163,921 | 165,398 | | 165,846 |
| Reference B8441 (clade I) | 949 | 1,022 | 970 | 962 | 958 | 961 | 922 | 64,316 | 44,605 | 165,846 | |

[a]The strains isolated from patients P0 and P1 are in boldface.

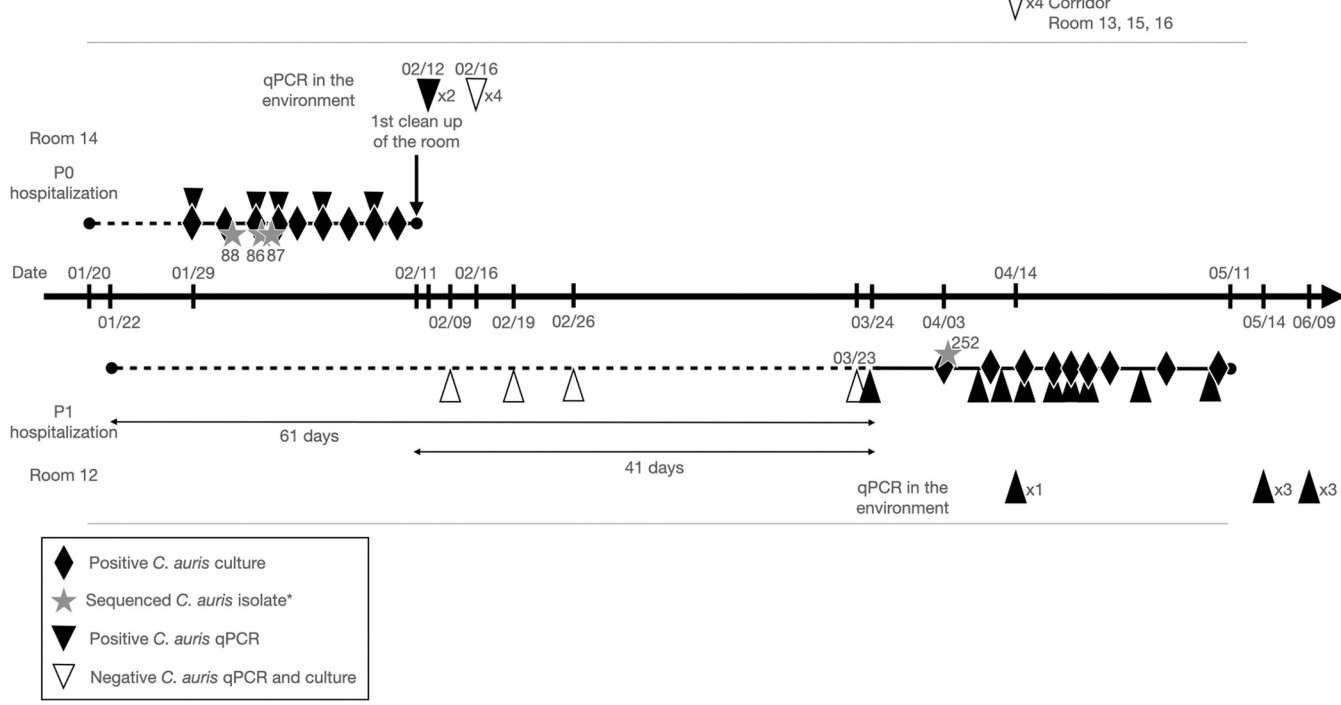

**FIG 1** Schematic representation of the P0 and P1 cases.

different between the 3 strains recovered from P0 was 6 (range, 0 to 6), suggesting that the patient carried one unique clade I strain. All *C. auris* qPCR results for all sites including nares, mouth, anus, and all other skin samples of the patient were thereafter positive in culture and qPCR (Fig. 1) (17). Finally, 13 days after the positive blood culture, P0 presented with a deadly bacterial septic shock due to *Escherichia coli* and *Pseudomonas aeruginosa*.

**Investigation of contact patients.** Contact patients were those patients present in the same ward at the time of the first positive sample of a detected patient. Seventeen contact patients during the hospitalization of P0 tested negative with qPCR and culture based on 3 weekly tests of axillary and groin swabs, as recommended (15).

However, one patient (P1) admitted just 1 day after the admission of P0, who was untested upon admission and negative upon screening (one test per week during 3 weeks) was determined to be positive for *C. auris* by qPCR and culture about 30 days after the last negative screening (Fig. 1). A phylogenetic tree constructed using the whole-genome SNP variants (Fig. 2) indicated that the isolate from P1 (CNRMA21-252) was genetically related to the three isolates from P0 (CNRMA21-86, CNRMA21-87, and CNRMA21-88) and distant from other clade I isolates from cases in France and reference strains, suggesting that P1 was infected with the strain from P0. The number of SNPs between the P0 and P1 strains was ≤12 SNPs (Table 1), supporting a very recent common ancestor and transmission. We estimated the incubation period for P1 to be between 41 days (between the last day P0 was present in the unit and first positive test of P1) and 61 days (between the first day P1 was in the unit and the first positive test of P1) (Fig. 1). P1 had *C. auris*-positive specimens from all skin (including nares) and digestive tract (mouth, anus) samples, without developing invasive infection. The antifungal susceptibility profile was the same as that of P0 (see Table S1), including a high MIC only to fluconazole. P1 was finally discharged from the hospital 109 days after admission, without invasive infection. The contact patients of P1 (*n* = 32) had three weekly negative axilla and groin swabs (culture and qPCR) on several skin sites (classical sites for colonization indexes). A total of 268 tests for 49 contact patients were performed and were negative.

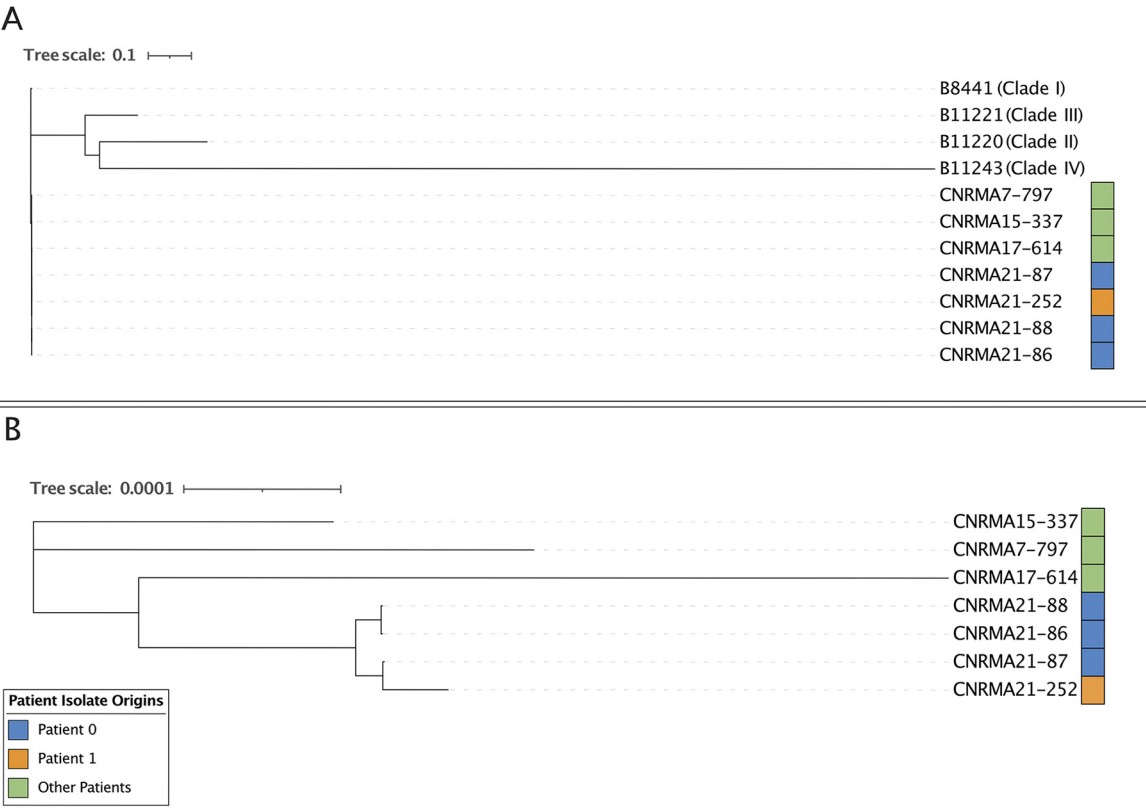

**FIG 2** (A) Maximum likelihood tree of the whole genome of the four isolates investigated in this study (CNRMA21-86, CNRMA21-87, CNRMA21-88, and CNRMA21-252), together with 3 other French clade I isolates and four of the *C. auris* clade reference genomes (B8441 [clade I], B11220 [clade II], B11221 [clade III], B11243 [clade IV]). (B) Maximum likelihood tree of the same four isolates investigated in this study, together with the 3 other French clade I isolates, but excluding the four clade reference genomes for clarity.

**Investigation of the environment.** *C. auris* detection in the environment after recommended biocleaning (double biocleaning with sodium hypochlorite [0.5%] and a sporicide [Incidin Oxyfoam, ANIOS Laboratories]) was positive, with qPCR only, on mattresses, the bed fence, and trolleys in the room of P0 (room 14) and P1 (room 12). Rooms 12 and 14 were separated by only one room on the same side of the corridor with an 8-m distance from each other. In P1's room, the detection of *C. auris* DNA was positive up to 1 month after discharge. Only a few shared materials between rooms were identified, as rooms were designed to be as independent as possible for each severe burn patient, allowing surgery within the room. The only shared material (an ultrasound device with probes) tested negative.

## DISCUSSION

We describe here the kinetics of colonization by *C. auris* of one patient in a burn ICU who acquired a clade I strain from another patient who was admitted most probably with *C. auris* colonization. It took more than 41 days for *C. auris* to be detected with P1 and to be finally detectable on all sites, despite three negative weekly screening tests. It is clear from the genetic analysis, which revealed a low number of SNPs between the P0 and P1 strains ($n < 12$), as already described for outbreak investigations (18), that the P0 strain had been transmitted to P1. Beyond this analysis, two likely scenarios coexist. (i) The contamination of P1 could have occurred at very early steps after admission, at a time when P0 was not known to be positive for *C. auris* (Fig. 1), either by a contaminated shared material or by health care worker hands, since no contact between patients was possible in the ICU. (ii) Some *C. auris* persisted in the environment, allowing P1 contamination either early or later on, leading to *C. auris*

detection after a period of 41 days. The latter seems less probable, as hygiene measures with an efficient *C. auris* cleaning solution had been extensively used in P0's room, including cleaning of shared material, and considering that all environmental samples came back negative in culture. In addition, no other contact patient was determined to be positive for *C. auris*, suggesting that no environmental source was significantly persisting.

We identified potential environmental sources of *C. auris*, such as the bed and its mattresses, but were never able to obtain a culture-positive environmental sample, suggesting that other routes of transmission were still plausible. These included transmission through the hands of the health care workers, as previously described (9, 12), through the sharing of material (echography machine), or very early before the first alert, all preventing finding the route of transmission. Indeed, it is possible that transmission occurred during the 7 days when both patients were hospitalized in parallel before the first recovery of *C. auris* in P0 (Fig. 1) and before an extensive and adapted cleaning procedure could be performed.

Using qPCR, we were able to detect DNA a long time after the discharge of patient P1 despite optimal cleanings and with persistently negative culture of these environmental specimens.

We never cultured *C. auris* from the environment in our settings, although DNA was detected frequently on the bed, the mattress, and the scope of the patient in both rooms of P0 and P1 up to 4 weeks after discharge of the patient and after several cleanings.

The use of qPCR clearly facilitated the management of the patients, because results could be obtained rapidly, preventing waiting for a positive culture plate, which would potentially need purification if the presence of a yeast mixture was detected before MALDI-TOF MS identification. Indeed, this qPCR assay not only can confirm *C. auris* species identification on colonies (because it has been designed and tested to be *C. auris* specific), but also can identify patients carrying *C. auris* on various body sites, with those results obtained within 24 h (17). Indeed, *C. auris* is known to grow slower than other yeasts (19), and so the identification of pink-white colonies on chromogenic medium (which could also be *C. parapsilosis* or other species) could be delayed. Daily observation of culture plates up to 7 days after inoculation seems important to detect compatible colonies (Fig. 3). We are aware that a new specific chromogenic medium can be implemented for this purpose, but in our case, we were not ready to purchase such a medium, for turnaround time reasons and because we were able to obtain results with qPCR testing within 24 h (20, 21).

Building on our experience, we then propose to use our *C. auris* qPCR assay to screen high-risk patients with an overnight stay in a health care facility from areas where *C. auris* and multidrug-resistant bacteria colonization and transmission are prevalent (Asia, Middle East, India and Pakistan, South and East Africa, Central and South America), according to CDC recommendations. Indeed, systematic screening of all new patients in the ICU is not an optimal strategy, as the prevalence of carriage has been shown to be very low (e.g., none of 998 admissions in ICUs in the London area) (22). Whether screening should be done only by qPCR rather than culture is still an open question and depends on the prevalence and on the turnaround time and costs that qPCR represents in each institution. For qPCR-positive contact patients, we recommend obtaining an isolate from at least one site to allow susceptibility testing on the isolates and to adapt the treatment accordingly (Fig. 3).

For known colonized patients, we propose to control the room cleaning with different specific surface swabs collected from various items of the room and waiting for negative *C. auris* culture results before admitting another patient into the potentially contaminated room (Fig. 3). Indeed, DNA detection can be positive a long time after a patient has left the room and despite several appropriate cleanings with persistent negative cultures, as the DNA detection could potentially correspond to persistent DNA of dead cells.

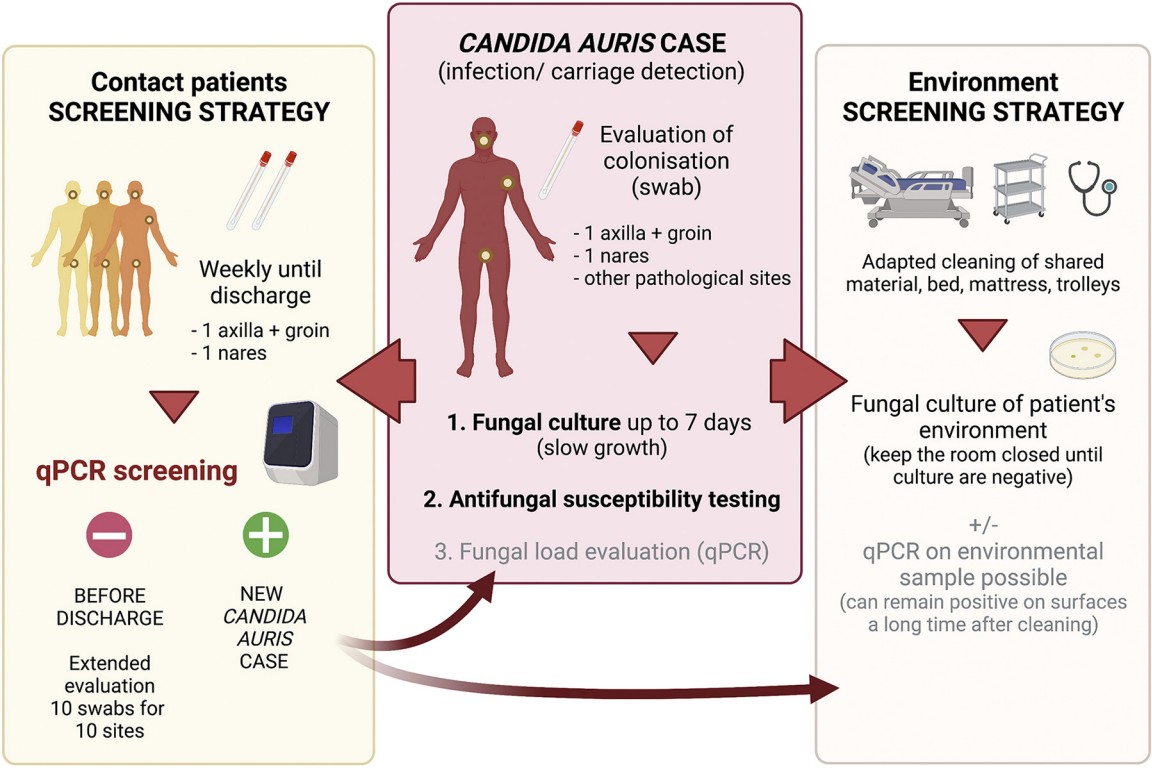

**FIG 3** Proposed recommendations for the management of *C. auris* cases based on our experience.

Finally, based on our experience, one should consider that transmission can be observed even late after exposure (>1 month) and can occur despite very separated and fully independent rooms and the implementation of control measures (isolation of the patients, rapid diagnosis) (Fig. 3). This example should prompt authorities to modify their recommendations by increasing the duration and the type of screening of patients at risk of transmission and at risk of developing invasive diseases, as suggested by our recent hospital guidelines (Assistance Publique-Hôpitaux de Paris) (see the supplemental material).

This investigation taught us several lessons. (i) Testing of patients coming from *C. auris* high-burden geographic areas should be done immediately upon admission to reinforce specific advertisements and measures to prevent *C. auris* transmission (hygiene measures here were appropriate in an optimal environment, including single rooms equipped with dedicated air treatment and a decontamination room). (ii) The screening strategy based on one test per week during 3 weeks of axilla and groin swabs may not be sensitive enough to detect a low fungal burden. Indeed, it has been demonstrated with clade IV isolates that the estimated sensitivity of such combined specimen testing is 60% (11). As this sensitivity reached 80 to 90% when additional sampling sites, including nares screening, were added (11), we recommend increasing the number of sites to be tested, including at least nares, even if we have not provided data in the present study. (iii) Negative screening should not exclude late colonization, with a maximum of 61 days from our experience. This should prompt authorities to modify their recommendations by increasing the duration and the type of screening of contact patients. (iv) *C. auris* can grow slowly for up to 7 days, especially if other yeasts are coexisting. Indeed, a qPCR-based screening strategy for contact patients is very efficient, as the result can be obtained in 24 to 48 h (20, 21), and should be recommended. Whether screening should be done only by qPCR rather than culture is still an open question and will depend on the prevalence and on the turnaround time and costs that qPCR represents in each institution. We argue that qPCR can be cost-effective, as

early results could avoid early transmission once *C. auris* is detected, saving diagnosis, hygiene, and potentially unneeded isolation procedures, but this should be evaluated in a specific study.

## MATERIALS AND METHODS

**Patient specimens.** Axillary and groin swabs from all patients involved in this investigation were collected in duplicates prospectively on Eswabs (Labelians, Nemours, France) and sent to our laboratory for qPCR analysis and culture.

**Classical mycology investigation.** Respiratory specimens of the *C. auris* patient were investigated with direct examination using calcofluor staining (BD Biosciences) in KOH (10%) and culture on BBL Chromagar (BD Biosciences) for 5 days at 35°C on malt agar extract (VWR) with gentamicin and chloramphenicol for 10 days at 30°C and Sabouraud dextrose agar with gentamicin and chloramphenicol (Bio-Rad) for 3 weeks. Of note, swab specimens were investigated with just the BBL Chromagar (BD Biosciences) for 5 days. White colonies on BBL were identified using the MALDI-TOF Bruker microFlex system (Bruker Daltonics, Bremen, Germany) with the MBT Compass IVD 4.2 database and the Vitek MS system (bioMérieux, Marcy l'Etoile, France) with the Vitek MS v1.6.0 database.

**Molecular identification.** Molecular identification was done based on sequencing of three different loci the internal transcribed spacer (ITS) locus using the V9D and LS266 primers, as already described (23). Sequences were submitted to the Mycobank database (http://www.mycobank.org/BioloMICSSequences.aspx?expandparm=f&file=all) and Institut Pasteur FungiBank (http://fungibank.pasteur.fr/).

**DNA detection.** All swab samples and other clinical specimens were extracted using the Qiasymphony DSP virus pathogen minikit (Qiagen) after bead beating with lysing matrix B (MP Biomedicals) in a Precellys bead beater (Bertin Technologies, Montigny-le-Bretonneux, France) and centrifugation in a Qiasymphony apparatus (Qiagen) with an elution volume of 85 $\mu$L (24). Eluates were then tested in duplicate using the *C. auris*-specific qPCR assay targeting 111 bp of the ITS locus (Cauris_Lima_262F, CGTGATGTCTTCTCACCAATCT, and Cauris_Lima_372R, TACCTGATTTGAGGCGACAAC) as previously reported, with replacement of the probe resulting in a new probe (Cauris_SLS_296P, 6-carboxyfluorescein [FAM]–TGCATTCACAAAATTACAGCTTGCACGAAA–black hole quencher 1) (25). Primer and probe concentrations were set at 0.3 and 0.1 $\mu$M in the 480 probe Master (Roche), respectively, and the PCR assay was performed in a 25-$\mu$L final volume (including 8 $\mu$L of eluate) at 58°C hybridization with 50 cycles of amplification in a LightCycler 480 instrument (Roche Life Science).

The results were expressed in quantification cycles ($Cq$'s), with higher values indicating less targeted DNA in the sample. Positivity was defined by at least one of the two duplicates having a $Cq$ of ≤45 cycles. DNA extraction and amplification yields were assessed using the Cy5 DiaControl DNA extraction and amplification internal control system (Diagenode, Seraing, Belgium), tested in a duplex qPCR with the target FAM assay.

As already evaluated by Lima et al., we first tested the specificity of the assay by testing 0.1 ng DNA from closely related strains (*C. auris* clade II CBS 10913, *C. auris* clade III CNRMA 20.272, *C. haemulonii* CBS 5149, *C. pseudohaemulonii* CBS 10004, *C. duobushaemulonii* CNRMA17.63, *C. lusitaniae* CBS 6936.) No amplification was observed in other strains or in species other than *C. auris*, confirming the specificity of the primers and probes. The PCR efficiency of the assay was 99% and the limit of detection was as low as 1 CFU (25).

Of note, the use of this assay for the detection of *C. auris* in a clinical specimen had been previously validated in our center based on specimens collected in a patient colonized with *C. auris* clade III, with culture-positive axillary and groin swabs. *C. auris* qPCR was positive in a culture-positive swab DNA extract, with a $C_q$ of 35.3 without bead beating versus 34.5 with bead beating.

**Environmental investigation.** Upon discharge and after biocleaning, sterile water-humidified cotton swabs were used to sample surfaces of the room. Upon arrival in the laboratory, swabs were unloaded in water. Culturing on Sabouraud dextrose agar (7 days) and *C. auris* qPCR were done as described above.

**Whole-genome sequencing.** Whole-genome sequencing was performed on 4 isolates from this study, CNRMA 21-088, CNRMA21-86, CNRMA21-87, and CNRMA21-252, on 3 clade I isolates from France (CNRMA7-797, CNRMA15-337, and CNRMA17-614), and on the clade I reference strain, B8441.

Genomes were sequenced at the Mutualized Platform for Microbiology (P2M, Institut Pasteur, Paris, France), using an Illumina NextSeq 500 sequencer. Libraries were constructed using a Nextera DNA library preparation kit and sequenced using a 2 × 150 nucleotide paired-end strategy.

**Computational analysis.** The *C. auris* assembly strain B8441 (GCA_002759435.2) was used as the reference genome for variant calling (26). Illumina sequences of additional clade I and other reference strains were generated in previous studies (3, 7). All isolate sequences were processed on Terra.bio. Isolate-paired FASTQs were processed into unmapped BAM files using the Terra workflow paired-fastq-to-unmapped-bam (https://portal.firecloud.org/?return=terra#methods/gatk/paired-fastq-to-unmapped-bam/10). The output unmapped BAM files were then run through fungal-variant-call-gatk4 (https://github.com/broadinstitute/fungal-wdl/tree/master/gatk4), which implements the GATK HaplotypeCaller for both SNPs and indels (27). Next, the per-sample GVCF files were combined and genotyped with CombineGVCFs and GenotypeGVCFs. Selected variants were filtered with VariantFiltration using "QD < 2.0 ‖ FS > 60.0 ‖ MQ < 40.0." Genotypes were filtered with a script in this workflow, requiring a minimum genotype quality of <50, percent alternate allele of <0.8, or depth of <10. The final variant calling format (VCF) file was annotated and given functional predictions using SnpEff (v. 4.3-t) (28), and also filtered for variants with a PASS flag using vcftools (v. 0.1.16) (29). To reduce the number of false-

positive identified variants, filtering of spanning deletions was also implemented by searching for variants with an alternative allele containing an asterisk.

We inferred a phylogeny to represent the relationships between each isolate. The VCF file was converted into FASTA format using a custom script (https://github.com/broadinstitute/broad-fungalgroup/blob/master/scripts/SNPs/vcfSnpsToFasta.py). Maximum likelihood phylogenies were built using RAxML v. 7.7.8 (30), with the GTRCAT nucleotide substitution model and 1,000 bootstrap replicates. To quantify the number of differences between isolates, a pairwise variant count matrix was built using a custom script (https://github.com/broadinstitute/broad-fungalgroup/blob/master/scripts/SNPs/fasta2snpcounts.pl). All phylogenies were visualized and annotated using iTOL (31).

**Ethics statement.** The patients were included in the PRONOBURN study protocol 2013/17NICB approved by the Institutional Review Board (00003835).

**Data availability.** Sequence data are available through NCBI Bioproject ID PRJNA865936.

## SUPPLEMENTAL MATERIAL

Supplemental material is available online only.

**SUPPLEMENTAL FILE 1**, PDF file, 0.9 MB.

## ACKNOWLEDGMENTS

We thank all the persons involved from the National Reference Center for Invasive Mycoses and Antifungals and the sequencing facility P2M platform of the Institut Pasteur. We also thank Corinne Maufrais at Institut Pasteur for her help on the genome analysis.

We declare no conflicts of interest.

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
