## [Reviewer comments · Microbiology Spectrum]

Microbiology Spectrum

First patient-to-patient intra-hospital transmission of a Clade I *C. auris* in France revealed after a two-months incubation period

Alexandre Alanio, Hannah Snell, Camille Cordier, Marie Desnos-Ollivier, Sarah Dellière, Samia Hamane, Aude Sturny-Leclère, Elodie Da Silva, Nesrine Aissaoui, Cyril Eblé, Martine Rouveau, Micheline Thégat, Widad Zebiche, Matthieu Lafaurie, Blandine Denis, Sophie Touratier, Mourad Benyamina, Emmanuel Dudoignon, Christina Cuomo, and Francois Dépret

Corresponding Author(s): Alexandre Alanio, Institut Pasteur

Review Timeline:

Submission Date:	July 15, 2022
Editorial Decision:	July 19, 2022
Revision Received:	July 26, 2022
Editorial Decision:	August 18, 2022
Revision Received:	August 25, 2022
Accepted:	August 25, 2022

Editor: Damian Krysan

Reviewer(s): The reviewers have opted to remain anonymous.

Transaction Report:

DOI: <https://doi.org/10.1128/spectrum.01833-22>

Prof. Alexandre Alanio
Institut Pasteur
Molecular Mycology Unit
28, rue du Dr Roux
Bat Duclaux aile Fourneau RDC Haut
Paris 75724
France

Re: Spectrum01833-22 (First patient-to-patient intra-hospital transmission of a Clade I C. auris in France revealed after a two-months incubation period)

Dear Prof. Alexandre Alanio:

I have reviewed of your manuscript entitled "First patient-to-patient intra-hospital transmission of a Clade I C. auris in France revealed after a two-months incubation period", and I regret to inform you that we will not be able to publish it in Spectrum.

Specifically, I have decided that the manuscript is outside of the scope for a microbiology journal such as Microbiology Spectrum and more appropriate for hospital epidemiology- or infection control-control focused journal. The findings are much more in line with these venues than with Spectrum.

I am sorry to convey a negative decision on this occasion, but I hope that the enclosed reviews are useful. Please note, rejections from Microbiology Spectrum are final and your manuscript will not be considered by other ASM journals. We wish you well in publishing this report in another journal and hope that you will consider Spectrum in the future.

Sincerely,

Damian Krysan
Editor, Microbiology Spectrum

Reviewer comments:

August 18, 2022

Prof. Alexandre Alanio
Institut Pasteur
Molecular Mycology Unit
28, rue du Dr Roux
Bat Duclaux aile Fourneau RDC Haut
Paris 75724
France

Re: Spectrum01833-22R1-A (First patient-to-patient intra-hospital transmission of a Clade I *C. auris* in France revealed after a two-months incubation period)

Dear Prof. Alexandre Alanio:

As you will read from the reviewers comments, there was some disagreement on the need for additional revisions. Reviewer 1 raises some important points regarding the strength of the conclusion that person-to-person transmission is the most likely mechanism. I think it would be best to soften this conclusion and provide a balanced commentary on alternate mechanisms in the revision. You may also want to soften a call for additional testing by suggesting that these approaches might provide reasonable approaches that could be studied in the future. The latter was also raised by reviewer 2. Careful attention to these comments as well as the other more minor issues is in order.

Link Not Available

Sincerely,

Damian Krysan

Journals Department
Reviewer comments:

Reviewer #1 (Comments for the Author):

The manuscript "First patient-to-patient intra-hospital transmission of a Clade I *C. auris* in France revealed after a two-months incubation period" by Alanio, et al., describes the first two patients with *C. auris* in a burn ICU. One of these patients had a positive blood culture and the other was colonized. The manuscript focuses on the kinetics of colonization of the second patient and argues that "person to person transmission" occurred but that the second patient had a prolonged incubation period prior to detection of colonization.

The authors present data that the second patient (P1) had a series of negative qPCR screening tests prior to becoming colonized. They report "It took more than 41 days for *C. auris* to significantly colonize the body of P1 and be finally detectable on all sites despite three negative weekly screening tests."

The limited number of SNPs between the two strains strongly supports the conclusion that the patients both had highly related strains that almost certainly had a common source. The first patient (P0) was mostly likely colonized on admission; this is the most likely source of the isolate.

The fact that the patients' isolates are related does not mean that there was "patient-to-patient" transmission of the organism. The authors cite many references and present their own data that *C. auris* persists on environmental surfaces for weeks to months. It can also be found on the hands of healthcare workers. The authors do not describe any direct contact between P0 and P1. It is far more likely that P1 acquired *C. auris* via nosocomial horizontal transmission related to environmental contamination with *C. auris*. This is by far the most frequently described mechanism of hospital-acquired *C. auris* and the simplest explanation for this transmission event. This mechanism of transmission is central to the manuscript because the authors conclude that P1 was colonized/infected with *C. auris* 41-61 days prior to P1's positive tests because that is the window of time when both P0 and P1 were hospitalized. There is no data to support a prolonged "incubation period" in P1. More likely, the environment was contaminated with *C. auris* from P0 and this contamination persisted long after P0 was discharged. P1 could have acquired colonization with *C. auris* from the environment at any time prior to P1's positive tests.

Other concerns include:

1. The manuscript still contains many phrases that are difficult to follow.
2. The authors refer to their institutional policies on *C. auris* surveillance in the text. These policies should be provided as supplemental material.
3. References 13-16 are various agency guidelines about *C. auris* infection prevention. Links to these guidelines or more specific information about where the reader can access the guidelines should be included. Reference 15 doesn't provide any source for the guideline.
4. The authors recommend that patients be screened at more sites than in their institutional guidelines and with culture techniques in addition to qPCR techniques without providing evidence to support this conclusion.

Reviewer #2 (Comments for the Author):

The authors have addressed the comments of prior review. The manuscript is easy to follow and interesting. I have the few suggestions for clarifications that would be helpful for the reader.

Were patients P0 and P1 negative for *C. auris* prior to the study, or untested?

Were other patients routinely screened on the floor/unit/building?

The authors suggest guideline changes may be needed. What guideline changes would they suggest and what type of cost may be associated?

Staff Comments:

Preparing Revision Guidelines

Please return the manuscript within 60 days; if you cannot complete the modification within this time period, please contact me. If you do not wish to modify the manuscript and prefer to submit it to another journal, please notify me of your decision immediately so that the manuscript may be formally withdrawn from consideration by Microbiology Spectrum.

Spectrum01833-22R2

Editor

As you will read from the reviewers comments, there was some disagreement on the need for additional revisions. Reviewer 1 raises some important points regarding the strength of the conclusion that person-to-person transmission is the most likely mechanism. I think it would be best to soften this conclusion and provide a balanced commentary on alternate mechanisms in the revision. You may also want to soften a call for additional testing by suggesting that these approaches might provide reasonable approaches that could be studied in the future. The latter was also raised by reviewer 2. Careful attention to these comments as well as the other more minor issues is in order.

Reply: Thank you for time and your comments. We responded to all reviewer's comments and modified the manuscript accordingly. This improved clearly the discussion. We hope that our manuscript will be accepted based on our revised version.

Sincerely,

Pr Alexandre Alanio

Reviewer #1 (Comments for the Author):

The manuscript "First patient-to-patient intra-hospital transmission of a Clade I *C. auris* in France revealed after a two-months incubation period" by Alanio, et al., describes the first two patients with *C. auris* in a burn ICU. One of these patients had a positive blood culture and the other was colonized. The manuscript focuses on the kinetics of colonization of the second patient and argues that "person to person transmission" occurred but that the second patient had a prolonged incubation period prior to detection of colonization.

The authors present data that the second patient (P1) had a series of negative qPCR screening tests prior to becoming colonized. They report "It took more than 41 days for *C. auris* to significantly colonize the body of P1 and be finally detectable on all sites despite three negative weekly screening tests."

The limited number of SNPs between the two strains strongly supports the conclusion that the patients both had highly related strains that almost certainly had a common source. The first patient (P0) was mostly likely colonized on admission; this is the most likely source of the isolate.

The fact that the patients' isolates are related does not mean that there was "patient-to-patient" transmission of the organism. The authors cite many references and present their own data that *C. auris* persists on environmental surfaces for weeks to months. It can also be found on the hands of healthcare workers. The authors do not describe any direct contact between P0 and P1.

It is far more likely that P1 acquired *C. auris* via nosocomial horizontal transmission related to environmental contamination with *C. auris*. This is by far the most frequently described mechanism of hospital-acquired *C. auris* and the simplest explanation for this transmission event.

This mechanism of transmission is central to the manuscript because the authors conclude that P1 was colonized/infected with *C. auris* 41-61 days prior to P1's positive tests because that is the window of time when both P0 and P1 were hospitalized. There is no data to support a prolonged "incubation period" in P1. More likely, the environment was contaminated with *C. auris* from P0

and this contamination persisted long after P0 was discharged. P1 could have acquired colonization with *C.auris* from the environment at any time prior to P1's positive tests.

Reply: We agree with reviewer that contamination from the environment is a very likely hypothesis. We were not clear enough in our manuscript.

Action: We modified the first chapter of the discussion to make Reviewer's hypothesis clearly stated and discussed line 149-171: " We describe here the kinetics of colonization to *C. auris* of one patient in a burn ICU who acquired Clade I strain from another patient who was admitted most probably with *C. auris* colonization. It took more than 41 days for *C. auris* to detect colonization of P1 and to be finally detectable on all sites despite three negative weekly screening tests. It is clear from the genetic analysis revealing a low number of SNPs between P0 and P1 strains ($n < 12$), as already described in outbreak investigations (17), that the P0 strain have been transmitted to P1. Beyond this analysis, two likely scenarios coexist: (i) the contamination of P1 could have occur at very early steps after admission, at the time where P0 was not known to be *C. auris*-positive (Figure 1) either by a contaminated shared material or by healthcare worker hands, since no contact between patients was possible in ICU ; (ii) Some *C. auris* persisted in the environment allowing P1 contamination either early or later on, leading to *C. auris* detection after a period of 41 days. The latter seems less probable, as hygiene measures using *C. auris* efficient cleaning solution have been extensively used in P0's room, including cleaning of shared material and considering that all environmental samples came back negative in culture. In addition, no other contact patient was detected positive, suggesting that no environmental source was significantly persisting.

We identified potential environmental sources of *C. auris* such as the bed and its mattresses but were never able to get a culture-positive environmental samples, suggesting that other routes of transmission were still plausible. These includes transmission through the hands of the healthcare workers, as previously shown (9, 12), through the sharing of material (echography machine) or very early before the first alert preventing finding the route of transmission. Indeed, it is possible that transmission could have occurred during the 7 days were both patients were hospitalized in parallel before the first recovery of *C. auris* in Patient 0 (Figure 1) and before an extensive and adapter cleaning procedure could be done".

Other concerns include:

1. The manuscript still contains many phrases that are difficult to follow.

Action: We shortened some phrases.

2. The authors refer to their institutional policies on *C.auris* surveillance in the text. These policies should be provided as supplemental material.

Action: We made available these guidelines as supplemental material 1.

3. References 13-16 are various agency guidelines about *C.auris* infection prevention. Links to these guidelines or more specific information about where the reader can access the guidelines should be included. Reference 15 doesn't provide any source for the guideline.

Action: We put hypertext links and modified references in the corresponding references.

4. The authors recommend that patients be screened at more sites than in their institutional guidelines and with culture techniques in addition to qPCR techniques without providing evidence to support this conclusion.

Reply: Screening nares in addition to groin and axilla are actually recommended in our local guidelines (not in the country ones).

Action: We chaged point (ii) line 211 as "(ii) The screening strategy based on one test per week during 3 weeks of axilla and groin swabs may not be sensitive enough to detect low fungal burden. Indeed, it has been demonstrated with clade IV isolates that the sensitivity of such combined specimen testing was estimated at 60% (11). As this sensitivity reach 80-90%, when additional sites including nare screening is added (11), we would recommend increasing the number of sites to be tested, including at least nares, even if we do not provide data in the present study. »

Reviewer #2 (Comments for the Author):

The authors have addressed the comments of prior review. The manuscript is easy to follow and interesting. I have the few suggestions for clarifications that would be helpful for the reader.

Were patients P0 and P1 negative for *C. auris* prior to the study, or untested?

**Action: We added line 86: “but was not tested for *C. auris* colonization upon admission”.
And line 116 “who was untested upon admission and negative upon screening”**

Were other patients routinely screened on the floor/unit/building?

Reply: As mentioned line 112 all the patients from the ward (n=17) were screening 3 times with culture and qPCR and were all negative.

Action: we change the sentence for “Seventeen contact patients during the hospitalization of P0 were tested negative with qPCR and culture upon 3 weekly testing of axillary and groin swabs as recommended [15].”

The authors suggest guideline changes may be needed. What guideline changes would they suggest and what type of cost may be associated?

Reply: line 209-210 we recommended to add nare screening in addition to groin and axilla and line 214-215, we recommended to use qPCR to obtaine reliable and early results.

Action: We indeed questioned costs and we are not able today to address this question. But this adding qPCR screening can be cost effective as the results come back early and that prevention procedures can be started earlier. To validate this cost effectiveness, a specific trial should be implemented. We added line XX: “We argue that qPCR can be cost-effective as early results could avoid early transmission when detected saving diagnosis, hygiene and potentially unneeded isolation procedures, but this should be evaluated in a specific study.”

August 25, 2022

Prof. Alexandre Alanio
Institut Pasteur
Molecular Mycology Unit
28, rue du Dr Roux
Bat Duclaux aile Fourneau RDC Haut
Paris 75724
France

Re: Spectrum01833-22R2 (First patient-to-patient intra-hospital transmission of a Clade I C. auris in France revealed after a two-months incubation period)

Dear Prof. Alexandre Alanio:

Thank you for your thorough response to the reviewer's comments and suggestions. I am pleased to accept it for publication.

Your manuscript has been accepted, and I am forwarding it to the ASM Journals Department for publication. You will be notified when your proofs are ready to be viewed.

Sincerely,

Damian Krysan
Editor, Microbiology Spectrum
